# Impeding Biofilm-Forming Mediated Methicillin-Resistant *Staphylococcus aureus* and Virulence Genes Using a Biosynthesized Silver Nanoparticles–Antibiotic Combination

**DOI:** 10.3390/biom15020266

**Published:** 2025-02-11

**Authors:** Mohamed A. Fareid, Gamal M. El-Sherbiny, Ahmed A. Askar, Amer M. Abdelaziz, Asmaa M. Hegazy, Rosilah Ab Aziz, Fatma A. Hamada

**Affiliations:** 1Clinical Laboratory Science Department, Applied Medical Science College, University of Ha’il, Hail 2440, Saudi Arabia; m.alekhtaby@uoh.edu.sa (M.A.F.); a.hegazy@uoh.edu.sa (A.M.H.); 2Botany and Microbiology Department, Faculty of Science, Al-Azhar University, Cairo 11884, Egypt; drahmed_askar@azhar.edu.eg (A.A.A.); amermorsy@azhar.eg (A.M.A.); 3Basic Sciences Department, First Year of Health and Medical Colleges, University of Ha’il, Hail 2440, Saudi Arabia; ro.abaziz@uoh.edu.sa (R.A.A.); f.hamada@uoh.edu.sa (F.A.H.)

**Keywords:** MRSA, Ag-NPs, antibiotics, biofilm, genes, synergistic effects, checkerboard assay, time-kill assay, cytotoxicity

## Abstract

Methicillin-resistant *Staphylococcus aureus* (MRSA) continues to represent a significant clinical challenge, characterized by consistently elevated rates of morbidity and mortality. Care regimen success is still difficult and necessitates assessing new antibiotics as well as supplemental services, including source control and searching for alternative approaches to combating it. Hence, we propose to synthesize silver nanoparticles (Ag-NPs) by employing a cell-free filter (CFF) of *Streptomyces* sp. to augment antibiotic activity and combat biofilm-forming MRSA. Seven bacterial isolates from clinical samples were identified, antibiotics were profiled with Vitek-2, and the phenotypic detecting of biofilm with Congo red medium and microplate assay was carried out. The PCR technique was used for detecting genes (*icaA* and *icaD*) coded in biofilm forming. The characterization of Ag-NPs was performed using several analytical methods, such as UV spectroscopy, dynamic light scattering (DLS), zeta potential measurement, transmission electron microscopy (TEM), X-ray diffraction (XRD), and Fourier transform infrared spectroscopy (FTIR). The antibacterial properties of Ag-NPs and oxacillin–Ag-NPs were assessed against standard strains and clinical isolates by employing the agar well diffusion technique and the microdilution assay. The biogenic synthesis Ag-NPs resulted in uniformly spherical particles, with an average size of 20 nm. These Ag-NPs demonstrated significant activity against biofilm-forming MRSA, with minimum inhibitory concentrations (MICs) ranging from 12 to 15 μg/mL. Additionally, Ag-NPs completely impede biofilm formation by MRSA at sublethal doses of 0.75 MICs. The expression levels of the *icaA* and *icaD* genes were reduced by 1.9- to 2.2- and 2.4- to 2.8-fold, respectively. A significant synergistic effect was noted when Ag-NPs were used in combination with oxacillin, leading to reduced MICs of 1.87 μg/mL for oxacillin and 4.0 μg/mL for Ag-NPs against MRSA. The FICi of 0.375 further validated the synergistic relationship between oxacillin and Ag-NPs at the concentrations of 1.87 and 4 μg/mL. Findings from the time-kill test demonstrated the highest reduction in log_10_ (CFU)/mL of the initial MRSA inoculum after 12-hour exposure. The cytotoxicity analysis of Ag-NPs revealed no significant cytotoxic effects on the human skin cell line HFB-4 at low concentrations, with IC_50_ values of 61.40 µg/mL for HFB-4 and 34.2 µg/mL for HepG-2. Comparable with oxacillin–Ag-NPs, Ag-NPs showed no cytotoxic effects on HFB-4 at different concentrations and exhibited an IC_50_ value of 31.2 against HepG-2-cells. In conclusion, the biosynthesis of Ag-NPs has demonstrated effective antibacterial activity against MRSA and has completely hindered biofilm formation, suggesting a valuable alternative for clinical applications.

## 1. Introduction

*Staphylococcus aureus* is a prevalent pathogen responsible for a diverse array of infections, ranging from minor dermatological issues to severe conditions such as endocarditis and osteoarticular infections, both of which are associated with significant morbidity and mortality rates [1]. Furthermore, methicillin-resistant *Staphylococcus aureus* (MRSA) is a significant and unpredictable threat, marked by its formidable and versatile nature. Its ability to adapt genetically and the ongoing emergence of successful epidemic strains ensure that it remains a critical concern for human health. In this context, *Staphylococcus aureus*, which is resistant to methicillin nowadays, has become a very dangerous pathogen, because it can thwart immune responses in several ways, giving rise to a high prevalence of both illness and death. Since its discovery in the 1960s, MRSA has developed a variety of protected mechanisms against antibiotic action, as well as ways to trick the host’s immune system [2,3]. Genomic studies indicate that the emergence of methicillin resistance occurred before the initial clinical application of anti-staphylococcal penicillins. This resistance is facilitated by the *mecA* gene, which is acquired through the horizontal transfer of a mobile genetic element known as the staphylococcal cassette chromosome mec (*SCCmec*). The *mecA* gene encodes penicillin-binding protein 2a (PBP2a), an enzyme that plays a crucial role in cross-linking the peptidoglycans within the bacterial cell wall. PBP2a exhibits a low affinity for β-lactams, which leads to resistance against this entire category of antibiotics [3].

The extracellular polymeric matrix of biofilms prevents antibiotic diffusion and encourages the emergence of multidrug-resistant communities. The primary contributors to biofilm formation in *Staphylococcus aureus* are twelve distinct genes. These include fibrinogen-binding proteins (fib), fibronectin-binding proteins (fnbA and fnbB), intercellular adhesion genes (*icaA*, B, C, and D), and clumping factor genes (*clfA* and B), in addition to elastin-binding proteins (ebps), laminin-binding proteins (eno), and collagen-binding proteins (cna). The intercellular adhesion genes *icaA, B, C, and D* play a crucial role in mediating cell-to-cell adhesion and the onset of biofilm formation. In addition, the co-expression of fnbA and B genes significantly contributes to the biofilm formation in *S. aureus*. Furthermore, it has been found that the fnbA and fnbB genes of *S. aureus* do not play a role in the adhesion process. Nevertheless, they facilitate the formation of biofilm; that is, fnA and B act as invasives, enabling *S. aureus* to enter the host cells [1,4,5,6,7,8,9].

Biofilm formation helps bacterial species overcome environmental conditions such as pH, temperature, tissue oxygen levels, and antibiotics that are not at their best. This makes MRSA capable of causing invasive illnesses, such as those involving the creation of biofilms. MRSA has proliferated rapidly, emerging as a pervasive bacterium responsible for numerous infections, owing to its diverse strategies for evading the host’s immune defenses. These infections, which can cause significant morbidity and mortality, range from recurrent, chronic skin and soft tissue infections (SSTIs) to deeper-seated ailments, including endocarditis and infections of the bones and joints (osteoarticular infections) [4,5,6]. Furthermore, the formation of bacterial biofilms on host tissues and medical instruments may promote several infections and make them challenging to eradicate. The biofilms encourage the persistence of infections because strains with increased virulence flourish in these barriers [7,9]. Compared to bacteria that live in biofilms, planktonic bacteria are 10–1000 times more susceptible to antimicrobial treatments. Based on information from the National Institute of Health, 80% of recurrent microbial infections are the result of bacteria associated with biofilm formation [10].

Treatment options that rely on nano-therapy are desperately needed, as the development of antibiotics has slowed down, and bacterial resistance has become more prevalent. Nanoparticles can aid in resolving these issues and improving the effectiveness of medications in this area [11,12,13]. Furthermore, nanoparticles possess the ability to selectively target certain biomolecules and microorganisms that hinder the formation of resistant strains. Furthermore, they exhibit efficacy as antimicrobial agents through distinct pathways in comparison to antibiotics [12,13,14]. The ability of nanoparticles to penetrate the cell membrane and cell wall of pathogenic microbes facilitates interference with essential molecular pathways, resulting in the formulation of innovative antimicrobial mechanisms. One effective strategy to combat growing bacterial resistance has been to combine antibiotics with metal nanoparticles to increase their activity [15,16].

The unique characteristics and extensive range of applications of silver nanoparticles have led to their significant recognition in the scientific community, particularly in the field of biomedicine [17]. The strong antibacterial and anticancer activities of these nanoparticles make them highly desirable for a wide variety of therapeutic applications [18]. Various studies have been conducted to investigate the antibacterial properties of Ag-NPs and their mechanisms of action. [19,20]. The antibacterial efficacy of silver ions (Ag^+^) is widely acknowledged, and silver nanoparticles exhibit greater antimicrobial activity than their bulk counterparts. The antimicrobial properties of Ag-NPs are primarily due to their ability to compromise the integrity of the plasma membrane, inhibit respiratory enzymes, and disrupt DNA replication processes [18]. Furthermore, silver nanoparticles have antibiofilm activity, according to Swidan et al. [21]. The green synthesis Ag-NPs exhibit antibiofilm activity greater than that from chemical synthesis against biofilm-forming *Staphylococcus aureus* [22]. Furthermore, the Ag-NPs obtained through green synthesis from *Artemisia scoporia* extract show significantly improved antibiofilm activity against multidrug-resistant *S. aureus* relative to that produced by chemical synthesis [23].

Furthermore, silver nanoparticles have recently been shown to have a synergistic effect and increase antimicrobial activity when coupled with multiple antibiotics, including ampicillin, amoxicillin, and chloramphenicol [24]. Conversely, research has indicated that Ag-NPs and amoxicillin or oxacillin interact antagonistically [20]. Investigations into the mechanisms of action of nanoparticle–antibiotic combinations have suggested that their increased antimicrobial efficacy may stem from chemical interactions. Nevertheless, additional clarification is required to uncover the fundamental molecular mechanisms at play, which could be either antagonistic or synergistic in nature [24]. Therefore, this study aims to biosynthesize silver nanoparticles, characterize them, and use them alone or develop a new combination with antibiotics to combat biofilm-forming MRSA.

## 2. Materials and Methods

### 2.1. Identification and Antibiotic Profiling of Bacterial Isolates

Seven bacterial isolates from skin wound infection swabs, identified as AS-1, AS-2, AS-3, AS-4, AS-5, AS-6, and AS-7, were collected from King Khalid Hospital in Ha’il, Saudi Arabia. These isolates were subsequently identified, and their antibiotic profiles were determined through the automated Vitek2 system (GP-card) Version 05.04, manufactured by BioMerieux SA, Marcy l’Etoile, France.

### 2.2. Determination of Biofilm Forming by MRSA Isolates

#### 2.2.1. Congo Red Agar (CRA)

Congo red agar medium consisting of (g/L) brain heart infusion (Difco) 37.0, sucrose 50.0, agar 10.0, and Congo red (Sigma-Aldrich, Taufkirchen Kreis München, Germany) solution 0.8 was used to qualitatively determine biofilm forming by MRSA isolates. The bacterial isolates were then transferred to plates with CRA medium and incubated for 24 h at 37 °C. After the incubation period, the bacteria that formed biofilms generated colonies that were black in color, while other bacteria resulted in red colonies [25].

#### 2.2.2. Microtiter Plate Assay (MPA)

The forming biofilms were assessed semi-quantitatively using 96-well microplates. Each isolate was cultivated in Trypticase Soy Broth (TSB) (Oxoid, Dublin, UK) comprising 1% glucose for 24 h at 37 °C. Diluted cultures were then added to the wells. Once the incubation period was over, the culture media and plates were treated with phosphate-buffered saline (PBS) at a pH of 7.2 to effectively eliminate any bacteria that were not adhered. The adherent biofilms were fixed, dried, and stained with crystal violet (Oxford, Dublin, UK). The stained biofilms were solubilized with ethanol at 95% and their optical density (OD) was assessed at a 492 nm wavelength using a plate reader in triplicate trials [21,26]. The classification of the bacteria that developed biofilms was conducted according to the data presented in Table 1.

#### 2.2.3. Detection Genes (*icaA* and *icaD*) Coded in Biofilm Formation

Utilizing the conventional polymerase chain reaction method, the presence of genes (*icaA* and *icaD*) coded in biofilm forming was detected. The National Centre for Biotechnology Information’s GenBank sequence database was used to validate the sequences of *icaA* and *icaD*. The EasyPure^®^ Bacteria Genomic DNA Extraction Kit (cat. no. EE161) (TransGen Biotech Co., Ltd., Beijing, China) and Solarbio^®^ Lysozyme (Lot. No. 928M045) (Beijing Solarbio Science & Technology Co., Ltd., Beijing, China) were employed for DNA extraction from MRSA strains and purification according to the manufacturer’s recommendations. The primers used for detecting these genes are recorded in Table 2. The thermocycling protocol consisted of the following steps: an initial denaturation for three minutes at 94 °C, followed by a series of cycles that included thirty seconds of denaturation at 94 °C, thirty seconds of annealing at 55 °C, and sixty seconds of extension at 72 °C. The protocol was completed with a final extension phase lasting two minutes at 72 °C [27]. The amplified DNA bands were seen with the use of UV light after being subjected to analysis using 1% agarose gel electrophoresis (Oxoid, Dublin, UK).

### 2.3. Biosynthesis of Silver Nanoparticles

The strain of *Streptomyces* sp. that was utilized to generate the sliver nanoparticles was previously isolated and identified [28]. For seven days at 28 °C, *Streptomyces* sp. was cultivated in a starch nitrate broth medium consisting of the following ingredients per liter: 2.0 g of KNO_3_, 20.0 g of starch, 0.5 g of MgSO_4_·7H_2_O, 1.0 g of K_2_HPO_4_, 0.5 g of NaCl, 3.0 g of CaCO_3_, and 0.01 g of FeSO_4_·7H_2_O, and incubated on an orbital shaker set at 150 rpm. Following the incubation period, cell debris was removed by centrifuging the cell-free supernatant for 30 min at 5000 rpm after filtering through cotton. The resulting filtrate was added with the 1 mM AgNO_3_ solution in a comparable ratio (vol./vol.). The reaction mixture’s color shift was employed to identify the synthesis of Ag-NPs. Both color observation and UV–vis spectrum measurement revealed the reduction in silver ions. After centrifuging and washing the resulting Ag-NPs in deionized water, they were dried in a hot air oven at 60 °C until a consistent weight was achieved [15].

### 2.4. Characterization of Biosynthesized Ag-NPs

The optoelectronic properties of the biosynthesized silver nanoparticles were evaluated using UV–visible absorption spectra (UV–vis, Hitachi U-2800, Hitachi High-Tech, Tokyo, Japan) covering the wavelength range of 300–700 nm. The quantification of nanoparticles can be achieved through UV–vis spectroscopy. By preparing five or more solutions of silver nanoparticles at varying concentrations, one can measure the absorbance of each sample. Subsequently, a graph can be constructed with concentration plotted on the *x*-axis and absorbance on the *y*-axis, leading to the derivation of a linear equation. This equation can then be utilized to ascertain the concentration of unknown samples. The zeta potential of the silver nanoparticles in phosphate-buffered saline (PBS) was determined with a Zetasizer Nano Particle Analyzer (Malvern Instruments Ltd., Worcestershire, UK). High-resolution transmission electron microscopy (HRTEM) (JEOL 2100, Akishima, Tokyo, Japan) was utilized at the National Research Center (NRC) in Giza, Egypt, to observe the size and shape of the nanoparticles. Furthermore, the particle size distribution of the Ag-NPs in phosphate-buffered saline (PBS) was assessed through dynamic light scattering (DLS) measurements using a Malvern Zetasizer instrument at the National Centre for Radiation Research and Technology (NCRRT) in Cairo, Egypt. The measurements were performed within a range of 0.1 to 1000 μm. To ascertain the crystalline characteristics of the Ag-NPs, X-ray diffraction (XRD) analysis was conducted using the Malvern Panalytical Empyrean 3 system from The Netherlands. The XRD analysis was executed at 40 kV and 30 mA, utilizing Cu Kα radiation with a wavelength of 1.54 Å, across a 2θ range of 10° to 90°. The average crystallite sizes of the Ag-NPs were calculated using the Scherrer equation. The Fourier transform infrared spectroscopy (FTIR) spectrum of the sample was recorded at the Chemistry Department, Faculty of Science, Al-Azhar University, Cairo, Egypt, employing an Agilent Cary 630 FTIR model to identify the functional groups present in the sample, with a comparison made against the reference chart and the collected spectral data [13].

### 2.5. Anti-Staphylococcal Activity of Biosynthesized Ag-NPs

A stock solution of Ag-NPs at a concentration of 5 mg/mL was prepared, and its antibacterial activity was assessed against the standard strain *S. aureus* ATCC 25923 and clinical MRSA isolates. The agar well diffusion method was applied, creating 8 mm wells in Mueller–Hinton agar plates (Merck Kga A, Darmstadt, Germany) that had been inoculated with the bacteria. Oxacillin disks (HiMedia Laboratories Pvt. Ltd., Mumbai, Maharashtra, India) were utilized as a positive control for this assay. A volume of 50 microliters of the Ag-NPs solution was pipetted into each well, followed by a 24-hour incubation at 37 °C [20,29]. Three duplicates of the experiment were carried out.

### 2.6. Evaluation of MICs of Oxacillin and Ag-NPs

Using a microdilution test in a 96-well plate, the MIC values of silver nanoparticles and oxacillin (HiMedia Laboratories Pvt. Ltd., Maharashtra, India) were assessed. A 10^6^ CFU/mL cell suspension of *S. aureus* ATCC 25923 and clinical isolates were used to inoculate Mueller–Hinton broth medium. One hundred microliters of the inoculated medium was added to each well. The substances that were tested, oxacillin and Ag-NPs, were diluted twice and tested again. Ag-NPs began at 100 to 1.0 µg/mL, whereas oxacillin was evaluated at doses ranging from 40 to 0.5 µg/mL. The experiment was conducted in accordance with the M7-A728 guidelines. To assess variations in optical density, wells containing negative controls (medium combined with oxacillin or Ag-NPs at the specified concentrations) were utilized. A volume of 30 µL of sterile resazurin solution (0.18%) from Hi-media was added to each well, followed by re-incubation at 35 °C for a duration of 18 to 24 h to evaluate the nutritional status of the bacterial cells. Visual inspection revealed that if the color of the microtiter plate wells remained blue or violet without alteration, the bacterial isolate was unable to grow at that concentration. Conversely, the color changes to pink, red, or purple indicated that bacterial growth was possible at that concentration. Also, the growth was monitored by assessing the absorbance at 600 nm with a microtiter enzyme-linked immunosorbent assay (ELISA) reader (Mindray MR-96A, Shenzhen, Guangdong, China). The lowest doses of silver nanoparticles or oxacillin that prevented the bacteria’s observable growth were identified as the MICs [30].

### 2.7. Antibiofilm Activity of Ag-NPs

After identifying the MICs for Ag-NPs, the values of 0.25, 0.5, and 0.75 MIC (sub-MIC) were applied to evaluate the antibiofilm potential of Ag-NPs against MRSA. The experiment was conducted following the protocol previously described in the crystal violet assay. Control wells included those with culture medium lacking Ag-NPs and those inoculated solely with the tested bacterial strains.

### 2.8. Expression of Genes (icaA and icaD) Coded in Biofilm Formation Before and After Treatment with Ag-NPs

A sub-MIC of Ag-NPs (0.75 MIC) (10.5 µg/mL) was prepared and introduced into a bacterial suspension at a 0.5 MacFarland concentration in triplicate. Bacterial isolates were not subjected to Ag-NPs treatment. The cultures were incubated at 37 °C for a duration of 24 h. The expression levels of the *icaA* and *icaD* genes were assessed through quantitative real-time PCR analysis, with the primer sequences detailed in Table 2. cDNA synthesis was carried out using the reverse transcriptase enzyme AMV (Roche, Basel, Switzerland) at a concentration of 25 units/L, in conjunction with transcriptase. To prevent the formation of secondary structures, the RNA extracted in the preceding step was heated to 65 °C for 3 min. Subsequently, reverse transcription (RT) was executed at 42 °C for 60 min, utilizing 2 μL of random primer and 2x AMV Reverse Transcriptase enzymes, followed by incubation and inactivation of the AMV enzyme at 99 °C for 5 min. Gene expression quantification was performed using the 2^−ΔΔCT^ method (sample CT minus control CT), where CT refers to the “threshold cycle” [31].

### 2.9. Assessment of Ag-NPs Combinations with Oxacillin Using Checkerboard Technique

A fresh stock solution of Ag-NPs was prepared at a concentration of 100 mg/mL, while oxacillin was formulated at a concentration of 10 mg/mL. The checkerboard assay was conducted using 96-well microplates containing Mueller–Hinton Broth (Difco, Georgia, USA). The Ag-NP concentrations were arranged from the minimum inhibitory concentration (MIC) to 1/128 MIC in the columns, and the oxacillin concentrations were organized from MIC to 1/128 MIC in the rows. The methicillin-resistant *Staphylococcus aureus* isolate, referred to as SA1, was introduced at a density of approximately 10^6^. Observations were recorded after 24-hour incubation at 37 °C. The control of growth was conducted exclusively with MRSA-containing media in wells that did not include combinations of oxacillin and Ag-NPs. For the negative control, wells were utilized that contained only the combinations used, without the presence of MRSA [32]. An ELISA plate reader, calibrated to 630 nm (Mindray MR-96A, Shenzhen, Guangdong, China), was employed to measure the optical density at both the commencement and conclusion of the experiment (after 24 h) to determine the variation in MRSA growth [33]. The relationship between oxacillin and Ag-NPs is elucidated through the fractional inhibitory concentration index (*FICi*), which can be calculated using the specified formula.$$FIC\;of\;oxacillin=\frac{MIC\;\;oxacillin\;in\;combination}{MIC\;\;oxacillin\;alone}\;\;\;\;$$$$FIC\;of\;\;Ag-NPs=\frac{MIC\;\;Ag-NPs\;in\;combination}{MIC\;\;Ag-NPs\;alone}$$

Therefore, the *FIC* index (*FICi*) is calculated as the sum of each *FIC*:FICi=FIC of oxacillin+FIC of  Ag−NPs

The interpretation of the results is as follows: a *FICi* of 0.5 or lower is categorized as synergy, while a *FICi* between 0.5 and 1 suggests partial synergy. A *FICi* from 1 to 2 is considered additive, values from 2 to 4 are classified as indifferent, and a *FICi* of 4 or more indicates antagonism [34].

### 2.10. Time-Kill Assay

The time-kill assay was employed to evaluate the bactericidal efficacy of oxacillin and silver nanoparticle (Ag-NPs) combinations, which demonstrated synergistic effects in the checkerboard assay against methicillin-resistant *Staphylococcus aureus* (MRSA) isolates. The MRSA cells were inoculated at a concentration of approximately 10^6^ CFU/mL in Mueller–Hinton broth, which was treated with oxacillin (1 µg/mL), Ag-NPs (8 µg/mL), various combinations of oxacillin and Ag-NPs at specified concentrations (0.5 MIC = (0.93 + 2) and 1 MIC = (1.87 + 4) µg/µg/mL, respectively), and untreated (control), as determined from the checkerboard results, and incubated at 37 °C for time intervals of 0, 1, 2, 3, 4, 5, 6, 7, 8, 10, and 12 h. Serial dilutions of the samples were performed in a sterile saline solution. Each dilution was inoculated in triplicate onto nutrient agar plates with a volume of 100 microliters. The measurement of colony-forming units per milliliter (CFU/mL) was conducted following an incubation period of 18 to 24 h at 37 °C [32,35,36]. The assessment of bacterial mortality rates and the measurement of lethality were conducted by plotting viable colony counts as log_10_ (CFU/mL) over time. It was established that the interaction could be classified as either bactericidal or bacteriostatic. A reduction of ≥3 log_10_ CFU/mL after 24 h of incubation was interpreted as a bactericidal effect, while a decrease of <3 log_10_ CFU/mL suggested a bacteriostatic effect [37].

### 2.11. In Vitro Cytotoxicity of Ag-NPs Against Normal and Cancer Cells

The cytotoxic effects of Ag-NPs and the combination of oxacillin–Ag-NPs (1:2 µg/µg) were evaluated in vitro using HFB-4 (normal human melanocyte) cell lines and HepG-2 (hepatocellular carcinoma) from ATCC, Rockville, with assessments conducted in triplicate as per the methodology outlined by El-Sherbiny et al. [12]. The MTT (3-(4,5-dimethyl thiazolyl-2)-2,5-diphenyltetrazolium bromide) assay was used to assess the proliferation potential and viability of the cells based on their metabolic activities. A medium containing varying quantities of Ag-NPs and oxacillin–Ag-NPs combinations (0.0, 1, 4, 8, 16, 32, 64, and 128 µg/mL) was substituted for the adherent culture medium, and the mixture was incubated for a whole day. After a 24-hour incubation, the cells were washed three times with fresh medium or cold PBS and then treated with 0.5 mg/mL MTT (Sigma-Aldrich) for 2 to 5 h. Once the MTT solution was discarded, 200 μL of DMSO was added to each well. The optical density (OD) of each sample was measured at 570 nm with a microplate reader (BMG LabTech, Offenburg,, Germany). The percentages of cell viability and death were determined using the following formulas.$$\%\;\mathrm{Cell}\;\mathrm{viability}=\frac{Treat\;cells}{Control\;cells}\times 100$$
$$\%\;\mathrm{Cell}\;\mathrm{death}=\frac{Control\;OD-Sample\;OD}{ControlOD}\times 100$$

### 2.12. Statistical Analysis

The results were presented as the mean ± SD value, which was determined using Microsoft Excel 365 and Minitab 18 software that had been supplemented with a statistical program.

## 3. Results and Discussion

### 3.1. Isolation, Identification, and Antibiotic Susceptibility of MRSA Isolates

With excellent probability ranging from 96% to 98%, we identified the clinical isolates as *Staphylococcus aureus* by employing the Vitek2 automated system, as shown in Table 3. The analysis of the susceptibility of these bacterial isolates to antibiotics was carried out using eighteen antibiotics from various classes. The findings demonstrated a considerable extent of multidrug resistance among the *Staphylococcus aureus* isolates examined. This study identified seven *S. aureus* isolates as methicillin-resistant *Staphylococcus aureus*, while these isolates continued to show sensitivity to amikacin, imipenem, and vancomycin, as illustrated in Table 4. Methicillin-resistant *Staphylococcus aureus* was initially discovered in clinical isolates from hospitalized patients in the 1960s, but since the 1990s, the population has seen a significant rise in its prevalence. MRSA is a formidable, versatile, unpredictable, and highly dominant pathogen of the modern era. It continues to pose a serious risk to human health because of its ability for genetic adaptability and the frequent appearance of effective epidemic strains [38]. The risk factors for MRSA infection often include extended hospital stays, admissions to intensive care units, recent hospitalizations, recent antibiotic use, MRSA colonization, invasive procedures, HIV infection, admission to assisted living facilities, open wounds, hemodialysis, and discharge with long-term central venous access or indwelling urinary catheters. Moreover, healthcare professionals who have direct contact with patients infected with MRSA are more likely to experience infections from this organism [39]. One of the most common causes of infections acquired in both hospitals and the community is multidrug-resistant *Staphylococcus aureus*, which is associated with high rates of morbidity and mortality. The difficulty of eradicating these infections is largely due to antibiotic resistance, and there is no effective vaccine available at this time [40]. Estimates from the Center for Disease Control and Prevention (CDC) indicate that more than 75,000 individuals in the United States contract an MRSA infection each year [39]. In light of the increasing prevalence of MRSA, vancomycin is regarded as the most effective treatment option for MRSA infections. The dense layers of the matrix present limited oxygen and nutrient availability, which contributes to the development of dormant persistent cells, thereby fostering antibiotic tolerance and resistance [41].

### 3.2. Extracellular Biofabrication and Characterization of Biosynthesized Ag-NPs

The challenge posed by multidrug-resistant bacterial infections, arising from developed resistance and/or biofilm, calls for the creation of innovative and unconventional drug therapies. In this regard, we are utilizing the cell-free filtrate of *Streptomyces* sp., combined in equal volumes with silver nitrate (1:1 *v*/*v*), as a reducing agent for the synthesis of silver nanoparticles. Previous studies have highlighted the potential of cell-free filtrates from *Streptomyces* species in the production of metal nanoparticles [20]. Abushiba et al. [30] conducted an investigation using cell-free filtrate from *Streptomyces rochei* cultivated in starch nitrate medium to synthesize silver nanoparticles. The biosynthesis of Ag-NPs was identified through the observation of a color change in the reaction mixture, which occurred when the precursor (AgNO_3_ solution, which appears white) was mixed with the yellow filtrate of *Streptomyces* sp., resulting in a dark brown coloration post incubation, as shown in Figure 1A. The observed color change in the reaction mixture can be linked to the reducing and capping agents present in the cell-free supernatant derived from *Streptomyces* sp.. The brown coloration of the Ag-NPs is attributed to the excitation of surface plasmon [40]. Since surface plasmon resonance (SPR) in metals underlies the development of color, the results were confirmed through UV–visible spectrum analysis to determine the most significant SPR peak in the reaction mixture. UV–visible absorbance spectroscopy is particularly advantageous for the analysis of metal nanoparticles, as the peak positions and shapes are sensitive to variations in particle size. A peak at 420 nm was noted in the UV–vis spectrum of the Ag-NPs when the AgNO_3_ solution was mixed with the filtrate of *Streptomyces* sp. during incubation, as shown in Figure 1A. This indicates the reduction of Ag^+^ to Ag^0^ by OH groups on cell-free filtrate of *Streptomyces* sp. This finding was in agreement with an investigation by Dayma et al. [42], who biosynthesized Ag-NPs using cell-free filtrate of *Streptomyces tendae* cultivated in malt glucose yeast peptone (MGYP) medium and silver nitrate as a precursor, which demonstrated UV–vis absorption spectra maxima at 420 nm.

Additionally, Elumalai et al. [43]’s biogenic Ag-NPs employed cell-free filtrate of actinobacteria from marine *Streptomyces coelicoflavus* MTK30. They showed that the UV–vis absorption spectra maxima were at 441 nm. Also, Nejad et al. [44]’s extracellular synthesis of silver nanoparticles employed cell-free filtrate of *Streptomyces* spp. Furthermore, Baygar et al. [45]’s research, utilizing silver nitrate as a precursor, revealed that the generation of silver nanoparticles (Ag-NPs) through the cell-free filtrate of *Streptomyces griseorubens* AU2, which was grown in ISP-2 broth medium comprising 4 g of yeast extract, 10 g of malt extract, and 4 g of dextrose, achieved peak absorption at 422 nm, confirming the existence of metallic Ag-NPs. The concentration of Ag-NPs was quantified at 2.70 × 10^8^ particles/mL. The zeta potential of these nanoparticles dissolved in PBS was −20.5 mV, and the conductivity was recorded at 0.0443 mS/cm, as shown in Figure 1B. The average particle size, distribution, and concentration of biosynthesized silver nanoparticles (Ag-NPs) were evaluated using nanoparticle monitoring analysis (NMA). This analysis relies on two key particle characteristics: light scattering and Brownian motion. In contrast to the Dynamic Light Scattering (DLS) method, the findings from the particle size analysis indicated a mean particle size of approximately 20 nm, as shown in Figure 1C. The present findings are consistent with the recent studies conducted by Dayma et al. and Vijayabharathi et al. [42,46]. The results indicated that the Ag-NPs generated by *Streptomyces griseoplanus* possess a negatively charged surface, with a zeta potential value recorded at −20.4 mV. This negative zeta potential value may result from the negatively charged functional groups located on the surface of the nanoparticles, which are derived from cellular metabolites [47]. Furthermore, the stability of nanoparticles in colloidal suspension is evidenced by the narrow and centered zeta potential [48]. These negatively charged nanoparticles are particularly advantageous for the formulation of drug–antibiotic complexes, which are crucial for various biological applications.

The analysis conducted using XRD enabled the identification of specific peaks that correspond to crystalline silver in the Ag-NPs produced through biological synthesis. The XRD spectrum of these biosynthesized Ag-NPs revealed notable diffraction patterns. Specifically, the peaks at 2θ values of 38.45˚, 46.35˚, 46.75˚, and 75.05˚ were found to correspond to the (111), (200), (220), and (311) crystallographic planes of Ag-NPs, respectively, as depicted in Figure 1D. These findings align with the earlier research by Sivasankar et al. [49], who reported that the XRD pattern of Ag-NPs synthesized from *Streptomyces olivaceus* (MSU3) reveals four significant peaks at 2θ values near 38.12°, 44.30°, 64.45°, and 77.41°. These peaks correspond to the lattice planes (111), (200), (220), and (311), respectively. Likewise, the study conducted by Dayma et al. [42] revealed five characteristic diffraction peaks that were indexed to the lattice planes (111), (200), (231), (222), and (220). In addition, Składanowski et al. [50] documented peaks at 2θ values of 38.1°, 44.6°, 64.6°, 77.5°, 81.5°, and 115.0° associated with crystalline Ag-NPs.

Also, the TEM image of the drying Ag-NPs revealed the spherical-shaped nanoparticles in dispersed form, as shown in Figure 1E. Recently, Elumalai et al. [43] successfully synthesized crystalline, spherical nanoparticles, measuring 98.4 nm in width, utilizing the cell-free filtrate of *Streptomyces coelicoflavus* MTK30. Furthermore, Nayka et al. discovered that *Streptomyces* sp. NS-33 produced spherical, polydisperse silver nanoparticles (Ag-NPs) with a size of 32.40 nm [51]. Additionally, Nejad [44], and their research team reported the formation of Ag-NPs from *Streptomyces* sp., exhibiting various hexagonal, pentagonal, spherical, and triangular shapes.

The FTIR study aimed to enhance the understanding of the functional groups associated with the synthesized Ag-NPs. The primary goal was to elucidate the conversion of these nanoparticles from simple inorganic silver salts (AgNO_3_) to their elemental form. Several metabolites, which function as reducing and capping agents, significantly influence this conversion process. In Figure 1Fa, the FTIR spectrum of CFF showed two peaks at 3353.77 and 2134.47, while the FTIR spectrum of Ag-NPs in Figure 1Fb reveals significant absorption peaks at 3353.77, 2134.47, 1638.91, 636.90, and 557.48 cm^−1^. The prominent peak at 3353.77 cm^−1^ is typically associated with hydroxyl functional groups, while the peak at 2134.47 cm^−1^ corresponds to CO_2_. The absorption at 1638.91 cm^−1^ is indicative of binding vibrations related to the protein amide I band, encompassing N-H stretching, suggesting the presence of functional groups linked to chemical bonds or organic compounds. This peak may also be associated with C-H bending vibrations in aliphatic compounds. The peak at 636.90 cm^−1^ generally reflects the stretching vibrations of C-O bonds, which are characteristic of various organic compounds, including esters, alcohols, or carboxylic acids, indicating the potential presence of these groups on the Ag-NPs. Lastly, the peak at 557.48 cm^−1^ often signifies the existence of metal–oxygen (M-O) bonds. It is important to note that the FTIR patterns found in this investigation closely matched those that Dayma et al. [42] reported. FT-IR analysis of Ag-NPs demonstrated bands in the region of 3328 to 531 cm^−1^. The stretching vibration of the alcohol (ROH) group and the C-O stretching mode are represented by the vibrations at 3328 and 2111 cm^−1^, respectively [52,53].

Furthermore, the peak at 1635 cm^−1^ in the FTIR spectrum was recognized as the C=O stretching vibrations associated with the amide linkages in proteins (amide II) detected in the cell-free supernatant [54]. The nanoparticles produced are stabilized and capped by the cell-free filtrate of *Streptomyces,* which inhibits their aggregation. The capped protein molecules may contain amino acid residues featuring hydroxy groups, as well as H_2_O molecules associated with the protein structure [55].

### 3.3. Antibacterial Activity of Ag-NPs

There is a growing interest in the revitalization of metal-based materials as viable alternatives to address the crisis of antibacterial resistance. Silver ions (Ag^+^) and silver nanoparticles have been employed as antimicrobial agents since ancient times and continue to find extensive application in the food industry and healthcare settings. Consequently, the antibacterial efficacy of Ag-NPs (suspended in sterile distilled water) against methicillin-resistant *Staphylococcus aureus* strains SA-1, SA-2, SA-3, SA-4, SA-6, SA-7, and *Staphylococcus aureus* ATCC 29523 was assessed using the agar well diffusion method. The findings of this study indicated that Ag-NPs demonstrate antibacterial properties against all MRSA isolates, with inhibition zones measuring between 12 ± 10 mm and 23 ± 10 mm, while the minimum inhibitory concentrations (MICs) ranged from 12 to 15 μg/mL, in contrast to oxacillin, which exhibited MICs of 16 to 32 μg/mL, as shown in Table 5 and Table 6. These findings align with an earlier investigation conducted by Sivasankaret al. [49], which demonstrated the high activity levels of Ag-NPs against Gram-positive bacteria, either standard or clinical strains, comparable with some antibiotics. In this context, the research conducted by Li et al. [56] revealed that silver nanoparticles demonstrate notable antibacterial activity, with MICs of 15 μg/mL against *S. aureus* and 50 μg/mL against MRSA. The antibacterial effect may be contingent upon the concentration of Ag^+^ ions released into the solution [57]. Silver nanoparticles are widely recognized as effective antimicrobial agents, exhibiting antibacterial properties that exceed those of some conventional antibiotics. Furthermore, they are utilized in wound dressings, implants, and catheters that are ready for clinical application [58].

We propose that the antibacterial mechanisms of silver nanoparticles involve their penetration into the cell wall, disruption of the plasma membrane, and direct interaction with DNA. This interaction leads to modifications that hinder DNA replication, as indicated by the inhibition of biofilm formation and the downregulation of genes related to biofilm production. A previous study was conducted by Sivasankar et al. [59], which demonstrated that the antibacterial mechanisms of silver nanoparticles are characterized by the following factors: (i) the penetration of Ag-NPs into the bacterial membrane, resulting in its destruction and the outflow of cellular contents; (ii) the production of reactive oxygen species (ROS) that interfere with respiratory processes; (iii) the modification of DNA or direct interaction with DNA, leading to alterations that affect DNA replication; and (iv) the denaturation of proteins and the inactivation of enzymes. Al-Wrafy et al. [58] have reported that the antibacterial properties of silver nanoparticles (Ag-NPs) are attributed to the dissolution of these nanoparticles and the subsequent release of silver ions. These ions integrate into the cell membrane, altering their permeability. Additionally, the negatively charged silver nanoparticles can infiltrate the target bacteria, leading to the degradation and oxidation of cellular components, disrupting respiratory chain enzymes, and generating reactive oxygen species (ROS). This process inhibits DNA replication and ATP synthesis. Notably, Ag-NPs possess a greater surface area-to-mass ratio compared to free Ag^+^, facilitating the regulation of their release kinetics and ensuring sustained antibacterial efficacy [60].

### 3.4. Biofilm Formation by MRSA

Infections caused by MRSA biofilms pose a significant therapeutic challenge. This study examined seven MRSA strains, of which five (71.4%) formed rough black colonies, as illustrated in Figure 2A. The remaining two strains (28.56%) were identified as non-producers, exhibiting smooth white colonies. Results from the microtiter plate assays revealed that among the seven strains tested, four (57.12%) were categorized as strong biofilm producers, two (28.56%) as moderate biofilm producers, and one (14.28%) as a poor biofilm producer. Polymerase chain reaction (PCR) analysis of the *icaA* and *icaD* genes in MRSA demonstrated that all isolates contained the *icaA* gene, measuring 188 base pairs. Additionally, the *icaD* gene was present in 6 MRSA strains, with a size of 198 base pairs; however, strain SA-5 was found to lack the *icaD* gene, as depicted in Table 7 and Figure 2B. Treatment of MRSA strains using Ag-NPs led to the total eradication of biofilm formation at sublethal doses of 0.75 MICs. In addition, the strains that underwent treatment with Ag-NPs became significantly more susceptible to antibiotics, as shown in Table 7. Figure 3 illustrates that the bacterial isolates of MRSA, when treated with Ag-NPs at a concentration of 10.5 µg/mL overnight, exhibited a reduction in the expression levels of the *icaA* and *icaD* genes by 1.9- to 2.2- and 2.4- to 2.8-fold, respectively, in comparison to the control isolates. This suggests that Ag-NPs lead to the downregulation of biofilm-associated genes, indicated by suppressed biofilm formation after treatment with Ag-NPs. Biofilms are recognized as significant contributors to the development of multidrug-resistant bacterial infections. According to reports from the National Institute of Health and the Center for Disease Control, bacteria that form biofilms account for 65–80% of all infections [10]. *Staphylococcus aureus* is capable of forming biofilms on both living and non-living surfaces, which significantly contributes to its widespread distribution and resistance to pharmaceuticals [61]. The cells that are surrounded and safeguarded by biofilms display distinct phenotypic characteristics when compared to their planktonic counterparts. *S. aureus* cells that have formed biofilms demonstrate a significant resistance to antibiotics and show variations in growth patterns, cell size, gene expression, and protein production relative to their free-living forms. Additionally, the process of bacterial biofilm formation is governed by specific genes associated with biofilm development [62].

The intercellular adhesion (ica) locus is responsible for the synthesis of polysaccharide intercellular adhesin (PIA), a crucial element in the biofilms of *Staphylococcus aureus*. The expression of the ica gene is linked to the process of biofilm formation. PIA is instrumental in biofilm development and is associated with infections related to biofilms, colonization, evasion of the immune response, and resistance to both antimicrobials and phagocytosis [63]. Therefore, the initial removal of biofilms is essential for achieving a more rapid and effective treatment. The degradation, removal, or even prevention of biofilm formation can be accomplished through various mechanisms and strategies. Among these, the interaction between nanoparticles and biofilm is influenced by factors such as the shape and size of the nanoparticles, the viscosity and charge of exopolysaccharides, cell density, compaction degree, liquid flow, and the physicochemical interactions with components of exopolysaccharides [64]. Hussain et al. [65] reported that silver nanoparticles (Ag-NPs) inhibited the formation of exopolysaccharides by 61–79% in Gram-negative bacteria and by 84% in Gram-positive bacteria. They linked this reduction in biofilm formation to the interference with quorum sensing, which is responsible for regulating alginate and exopolysaccharide production by hindering the bacteria’s ability to adhere to surfaces, due to the impact of Ag-NPs on bacterial motility. Lewis Oscar et al. [66] illustrated that silver nanoparticles inhibited biofilm formation by *Pseudomonas aeruginosa* by interfering with the production of rhamnolipid through the modulation of the quorum sensing mechanism. Furthermore, silver nanoparticles interfere with protein functionality by attaching silver ions to cysteine residues, resulting in alterations to protein structures and impeding the production of exopolysaccharides [67]. Giri et al. [68] demonstrated that the biosynthesis of Ag-NPs effectively eliminates both biofilm formation and the biofilm-producing strain of *Staphylococcus aureus*.

Nanoparticles are emerging as a novel strategy for the elimination of persistent multidrug-resistant planktonic bacterial and biofilm infections [41]. Ongoing in vivo and in vitro research aims to assess the potential of this method for use in wound dressings. Additionally, an amphiphilic core–shell polymeric nanomaterial has been developed, which effectively removes preformed biofilms of MRSA through nanoscale bacterial debridement [69]. An in vivo mouse excisional wound biofilm model demonstrated successful dispersal of methicillin-resistant *Staphylococcus aureus* biofilms [70]. In this investigation, the level of oxacillin resistance increased significantly over 16 serial passages, with the minimum inhibitory concentration (MIC) rising dramatically by a factor of 16. Wang et al. also reported a significant elevation in the resistance level to ampicillin over 16 serial passages, with the MIC increasing by 16-fold [71].

### 3.5. Checkerboard Assay

Using the checkerboard microdilution approach, several combinations of oxacillin and Ag-NPs have been developed with the goal of enhancing their activity and decreasing their dose. Testing the antibacterial effectiveness of these combinations was performed on test strain MRSA. The conventional checkerboard technique serves as a general procedure for determining synergy. It involves evaluating two medications at various concentrations and combinations in order to determine the concentration of each ingredient that has the least amount of synergistic effect on the other. The fractional inhibitory concentration index (FICi) can be utilized for calculating the interaction algebraically. The degree of interaction between oxacillin and Ag-NPs against MRSA is indicated by the FICi. In this context, when combinations of oxacillin and Ag-NPs were evaluated against methicillin-resistant *Staphylococcus aureus*, a total of 28 treatments were produced, each varying in their degree of interaction. Among these, fifteen demonstrated indifferent interactions, ten displayed additive interactions, while only three exhibited synergistic effects, with fractional inhibitory concentration indices (FICi) of 0.5 and 0.375, as shown in Table 8 and Figure 4. The (FICi) result was consistent with data obtained by Wang et al. [71] when using the checkerboard microdilution assay, observing a notable synergistic effect between silver ions or silver nanoparticles and various antibiotics, including kanamycin, ampicillin, ciprofloxacin, and tetracycline, against *S. aureus* Newman, with FIC index values (FICi) ranging from 0.1875 to 0.375.

In this study, the oxacillin and Ag-NPs, when combined with each other, were enhanced by one another, leading to a decrease in both of their MIC values when compared to each of them alone. The interactions resulted in a reduction in the MIC of oxacillin from 32 µg/mL to below 4 µg/mL, representing an eightfold decrease. Similarly, the MIC for Ag-NPs decreased from 15 µg/mL to under 1.87 µg/mL, also indicating an eightfold reduction. Several decades earlier, the most recommended treatment for infectious diseases was monotherapy. However, there is currently sufficient evidence to demonstrate that multitherapy or combination antimicrobials are more effective than single-drug therapies. One popular method for determining the effectiveness of combined antimicrobials is the checkerboard broth microdilution test. The outcomes are used to demonstrate the additive, antagonistic, synergistic, or indifferent effects of the antimicrobials [13]. The results of the oxacillin and Ag-NPs test combined clearly show that the checkerboard assay determined each agent’s concentration in the combination when the FIC index was less than 0.5. By lowering the dosage of each agent to at least one-eighth of the respective MIC, this synergistic combination of oxacillin and Ag-NPs may decrease toxicity while enhancing antibacterial activity. When combining antibiotics with NPs, they are more effective against drug-resistant bacteria, demonstrating the synergistic effects of Ag-NPs in combination with tetracycline, cefixime, ciprofloxacin, and bacitracin against *E. coli*, *P. aeruginosa*, *S. aureus*, and *Candida albicans* [72].

The synergistic effects of oxacillin, azithromycin, cefotaxime, cefuroxime, oxytetracycline, and fosfomycin have shown greater efficacy against *S. aureus* and *E. coli* compared to the use of each drug individually. This enhanced effectiveness may be attributed to the oxidative stress induced by the nanoparticles on the cell surface, which resulted in increased permeability [73]. The use of nanocarriers can diminish the development of resistance by delivering drugs that activate various effective mechanisms, while also enabling the targeted release of substances, thereby preventing bacteria from being exposed to sublethal drug concentrations [74]. In this regard, multidrug-resistant *E. coli* isolates can be effectively treated with levofloxacin that is encapsulated within silver core-embedded mesoporous silica nanoparticles (Ag@MSNs@LEVO). The combination of levofloxacin and Ag@MSNs exhibited a synergistic antibacterial effect. The silver component not only functioned as a carrier, but also provided antimicrobial activity through the generation of silver ions. Administration of Ag@MSNs@LEVO resulted in a three-log reduction in bacterial load in an in vivo mouse peritonitis model, leading to decreased damage to the spleen and peritoneum, with no toxicity detected [75]. In a similar approach, ampicillin was conjugated to the surface of Ag-NPs, resulting in broad-spectrum bactericidal agents capable of circumventing the resistance mechanisms present in multidrug-resistant strains of MRSA, *Enterobacter aerogenes*, and *Pseudomonas aeruginosa* [76]. Generally, nanomaterials demonstrate various bactericidal pathways to combat bacterial infections and address antibiotic resistance. The bioengineering of their shape, size, and surface properties offers extensive opportunities for the development of novel antimicrobial agents [41].

### 3.6. Time-Kill Assay

The killing kinetic technique was employed to measure the bare minimum time for combinations (oxacillin–Ag-NPs) to achieve a bactericidal or inhibitory effect on the viability of bacteria cells after treatment. MRSA was treated with oxacillin 1 μg/mL, Ag-NPs 8 μg/mL, the oxacillin–Ag-NPs combination at concentrations (oxacillin–Ag-NPs, 0.5 MIC = (2 + 0.93) or 1MIC = (4 + 1.875) μg/μg/mL, respectively), and untreated (control), which had been calculated from the checkerboard test and different time intervals 0, 1, 2, 3, 4, 5, 6, 7, 8, 10, and 12 h, as shown in Figure 5A,B. The time-kill assay demonstrated the effectiveness of oxacillin–Ag-NPs in suppressing and killing the bacterial species in a dose- and time-dependent manner. Oxacillin–Ag-NPs have bactericidal action against bacteria isolates; the number of CFU/mL is significantly decreased when compared to the growth of the controls (untreated), with an ascending growth curve during the entire test period. The bactericidal endpoint of oxacillin–Ag-NPs for methicillin-resistant *Staphylococcus aureus* isolates SA1 was reached after 2 h of incubation at 1MIC (1.87 + 4 µg/µg/mL), while for isolate SA5, the bacteria was killed after 3 h of incubation at same concentration. In this regard, the bactericidal endpoints of oxacillin–Ag-NPs for two bacterial isolates were reached after 6 h of incubation at a concentration of 0.5 MIC (0.93 + 2 µg/µg/mL).

The use of combination drugs has been shown to significantly inhibit the development of highly resistant bacterial strains. Research indicates that silver ions or silver nanoparticles, in conjunction with ampicillin, exhibit a synergistic effect against MRSA 33591. Specifically, the introduction of 8 μg/mL of Ag^+^ or Ag-NPs can lower the minimum inhibitory concentration (MIC) of ampicillin by a factor of 16. Additionally, time-kill assays revealed that the combination of Ag^+^/Ag-NPs and ampicillin produced a markedly enhanced bactericidal effect, resulting in a 6–8-log_10_ reduction in the viability of *S. aureus* over a 24-hour period, compared to control or monotherapy. This suggests that Ag^+^/Ag-NPs have the potential to resensitize MRSA to antibiotic treatment [71]. In this scenario, the multi-target mode of action exhibited by Ag-NPs imparts a sustainable antibacterial capability, which not only enhances the effectiveness of standard antibiotics, but also re-sensitizes MRSA to these drugs [71]. Despite numerous research endeavors, the antibacterial mechanisms of Ag^+^ against *S. aureus* are still largely unclear. To date, significant research has concentrated on the proteins that are up- and down-regulated in *S. aureus* following exposure to Ag-NPs or Ag^+^ through quantitative proteomic approaches [77,78]. The Ag^+^ transcriptome within *Staphylococcus aureus* is analyzed to elucidate the dynamic bactericidal mechanism of action of antibacterial Ag^+^ against this pathogen at the molecular level. Silver ions demonstrate bactericidal effectiveness by interfering with numerous biological pathways, leading to the functional disruption of essential proteins. This multifaceted approach to action allows Ag^+^ to maintain a prolonged antimicrobial effect [71]. A previous investigation has shown that silver ions and Ag-NPs can enhance the effectiveness of a diverse array of antimicrobials, restore the sensitivity of MRSA to antibiotics, and reduce antibiotic resistance in *Staphylococcus aureus*. As a result, the combination of antibiotics with silver or other metal-based substances, such as colloidal bismuth sub-citrate (CBS) or auranofin, may offer a promising method to reduce the selective pressure of antibiotics, thereby preventing the emergence of primary antibiotic resistance and extending the lifespan of traditional antibiotics to combat the ongoing crisis of antibiotic resistance [79,80].

### 3.7. In Vitro Cytotoxicity of Biosynthesized Ag-NPs and Their Combination Against Normal and Cancer Cells

The potential clinical application of Ag-NPs and their combination oxacillin–Ag-NPs as antibacterial agents must be considered so that it is not harmful to normal cells. As depicted in Figure 6A, Ag-NPs demonstrated no toxicity to normal cells at low concentrations, suggesting their potential as non-toxic drug delivery agents at concentrations of 8 µg/mL or lower. The IC_50_ values were recorded at 61.40 µg/mL and 34.2 µg/mL for HFB-4 and HepG-2 cells, respectively. In comparison, oxacillin–Ag-NPs exhibited no significant cytotoxic effects on normal human cells (HFB-4) across a concentration range from 100 to 10.0, indicating their safety for use with normal human cells. Furthermore, the combination of oxacillin–Ag-NPs showed anticancer activity against HepG-2 cells, with an IC_50_ value of 31.2 µg/mL, as shown in Figure 6B. The findings indicate that the IC_50_ value of oxacillin–Ag-NPs against HepG-2 cells is 36.4 µg/mL. A recent study conducted by Abdel-Fatah et al. [59] revealed that Ag-NPs synthesized using an aqueous extract of *Carthamus tenuis* exhibited no cytotoxic effects on normal human cells (HFB-4) across a range of concentrations from 100 to 1.56 µg/mL. The authors concluded that Ag-NPs are safe for normal human cells while demonstrating anticancer activity against HepG-2 cells, with an IC_50_ value of 5.6 µg/mL. Furthermore, biologically synthesized Ag-NPs displayed significant cytotoxic effects on MCF-7 and HT-29 cell lines [81]. The human body has the capacity to tolerate a daily oral ingestion of silver nanoparticles ranging from 0.4 to 27 μg [82]. The research indicates that human cells exhibit significant resistance to the toxic effects of Ag-NPs when compared to other cell types [83]. Additionally, biogenically synthesized Ag-NPs are found to be biocompatible, posing no harm to normally functioning human or host cells [84].

## 4. Conclusions

Infections caused by *Staphylococcus aureus*, particularly those associated with biofilm formation and acquired resistance, necessitate the exploration of innovative drug development strategies. Various analytical techniques, including UV–visible spectroscopy, DSL, zeta potential measurement, TEM, XRD, and FT-IR, have validated the successful synthesis of pure and stabilized Ag-NPs using a cell-free filter method. The biogenic Ag-NPs demonstrated notable antibacterial efficacy against clinical strains of MRSA and effectively inhibited biofilm development via the downregulation of their associated genes. Additionally, the combination of Ag-NPs with oxacillin significantly enhanced antibiotics’ effectiveness against MRSA clinical isolates under the experimental conditions of this study. Furthermore, Ag-NPs represent a burgeoning trend in bio-nanotechnology, offering extensive applications in the biomedical sector due to their environmentally friendly properties and biocompatibility. Our results strongly advocate the use of Ag-NPs as innovative and advanced solutions for addressing MRSA-related planktonic and biofilm infections. A comprehensive evaluation of the clinical application of biosynthesized silver nanoparticles in combination with oxacillin requires the implementation of a well-controlled in vivo and clinical trial.

## Figures and Tables

**Figure 1 biomolecules-15-00266-f001:**
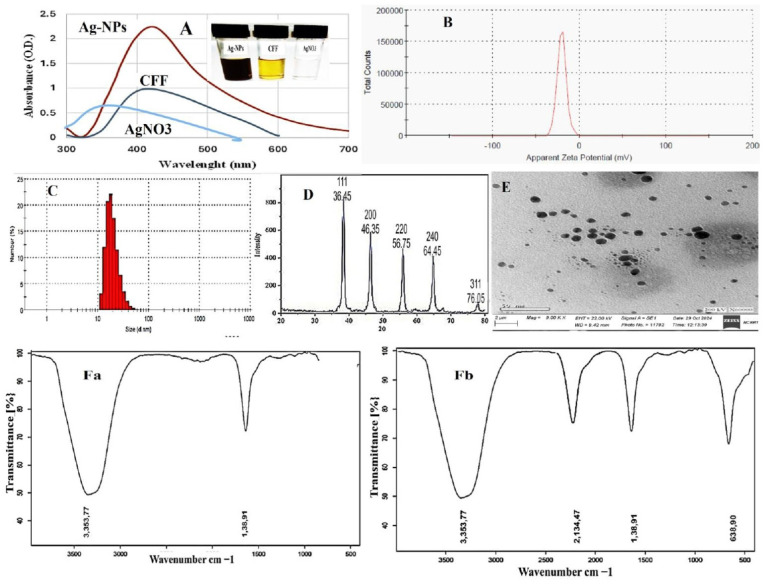
Biosynthesized and characterized Ag-NPs, (**A**) UV–visible, (**B**) zeta potential, (**C**) DLS pattern, (**D**) X-ray diffraction pattern (XRD), (**E**) transmission electron microscopic (TEM), (**Fa**) FTIR spectrum of CFF, and (**Fb**) FTIR spectrum of Ag-NPs.

**Figure 2 biomolecules-15-00266-f002:**
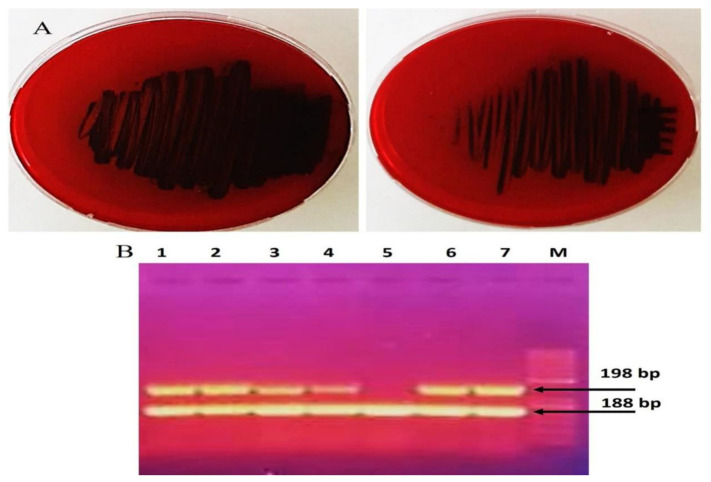
Detection of biofilm formation by MRSA strains (**A**) phenotypic with Congo red and (**B**) genotypic with detect *icaA* and *icaD* genes (Original gel electrophoresis images can be found in Appendix A).

**Figure 3 biomolecules-15-00266-f003:**
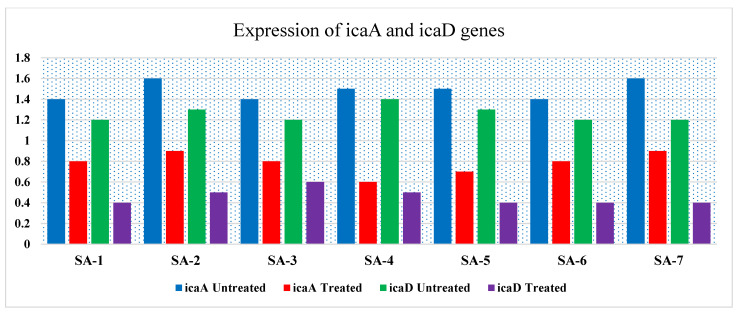
RT-PCR expression of the biofilm-associated genes (*icaA* and *icaD*) treated with Ag-NPs, sub-MIC level and untreated (control).

**Figure 4 biomolecules-15-00266-f004:**
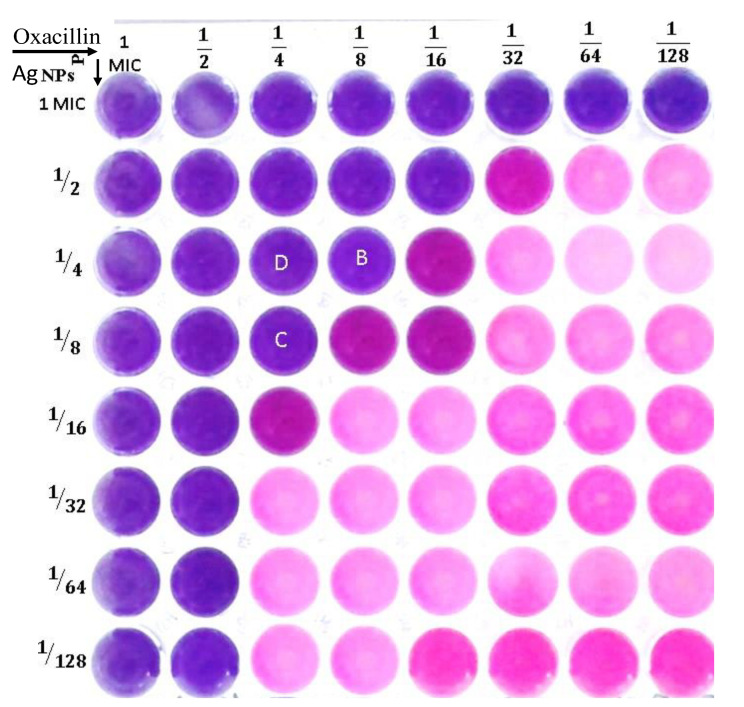
Checkerboard assay to test the synergistic interaction of combined oxacillin and Ag-NPs against methicillin-resistant *S. aureus*.

**Figure 5 biomolecules-15-00266-f005:**
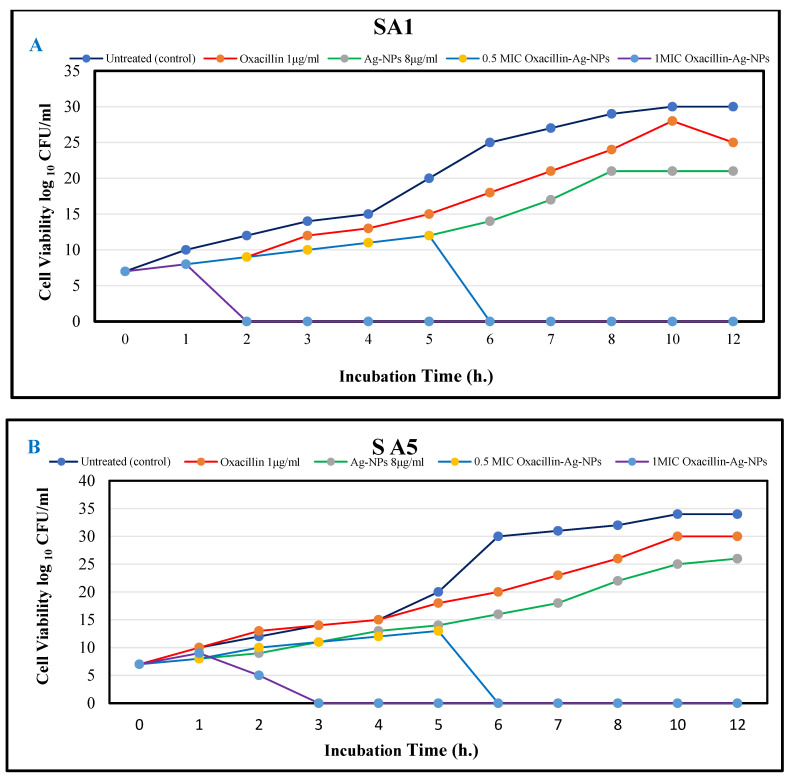
Time-kill assay of oxacillin, Ag-NPs, and oxacillin–Ag-NPs against (**A**) SA1 and (**B**) SA5.

**Figure 6 biomolecules-15-00266-f006:**
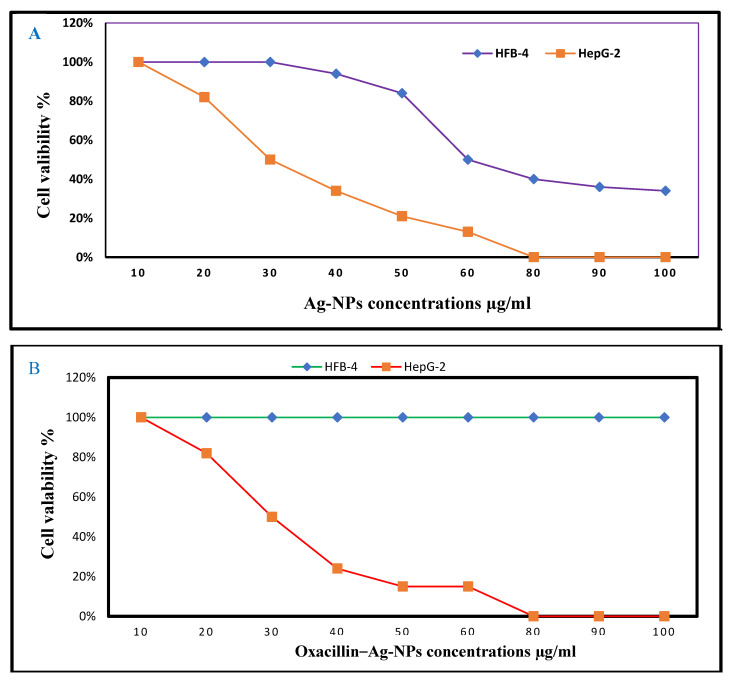
Dose–response curves of cytotoxic activity of (**A**) Ag-NPs and (**B**) oxacillin–Ag-NPs on the viability of HFB-4 and HepG2 cell lines.

**Table 1 biomolecules-15-00266-t001:** Bacteria categorized based on their capacity for biofilm formation.

Results	Biofilm Class
Strong biofilm	OD > 4 × ODc
Medium biofilm	2 × ODc < OD ≤ 4 × ODc
Poor biofilm	ODc < OD ≤ 2 × ODc
Negative biofilm	OD ≤ ODc

**Table 2 biomolecules-15-00266-t002:** Primers used in the study.

Target Gene	Primers	Melting Temperature—Tm (°C)	Product Size (bp)	Ref.
*icaA*	F 5′ TCTCTTGCAGGAGCAATCAA 3′ R 5′ TCAGGCACTAACATCCAGCA 3′	55.5	188	[27]
*icaD*	F 5′ ATGGTCAAGCCCAGACAGAG 3 ′ R 5′ CGTGTTTTCAACATTTAATGCAA3′	55.5	198	[27]

**Table 3 biomolecules-15-00266-t003:** Identification of the methicillin-resistant isolates by Vitek2 system.

Isolate Code	Name of Bacteria	Confidence Level	% Probability
AS-1	*Staphylococcus aureus*	Excellent	98%
AS-2	*Staphylococcus aureus*	Excellent	98%
AS-3	*Staphylococcus aureus*	Excellent	98%
AS-4	*Staphylococcus aureus*	Excellent	98%
AS-5	*Staphylococcus aureus*	Excellent	96%
AS-6	*Staphylococcus aureus*	Excellent	96%
AS-7	*Staphylococcus aureus*	Excellent	98%

**Table 4 biomolecules-15-00266-t004:** Antibiotic susceptibility of methicillin-resistant *Staphylococcus aureus* isolates tested.

Antibiotics	AS-1		AS-2		AS-3		AS-4		AS-5		AS-6		AS-7	
	MIC	Profile	MIC	Profile	MIC	Profile	MIC	Profile	MIC	Profile	MIC	Profile	MIC	Profile
Methicillin	≥16	R	≥16	R	≥16	R	≥16	R	≥16	R	≥16	R	≥16	R
Amoxicillin	≥32	R	≥16	R	≥32	R	≥32	R	≥32	R	≥32	R	≥32	R
Oxacillin	>4	R	>4	R	>4	R	>4	R	>4	R	>4	R	>4	R
Nalidixic acid	≥64	R	≥64	R	≥64	R	≥64	R	≥64	R	≥64	R	≥64	R
Ciprofloxacin	≥64	R	≥64	R	≥64	R	≥64	R	≥64	R	≥64	R	≥64	R
Rifampin	>4	R	>4	R	>4	R	>4	R	>4	R	>4	R	>4	R
Amikacin	≤16	S	≤16	S	≤16	S	≤16	S	≤16	S	≤16	S	≤16	S
Nitrofurantoin	≥16	R	≥16	R	≥16	R	≥16	R	≥16	R	≥16	R	≥16	R
Clindamycin	>4	R	>4	R	>4	R	>4	R	>4	R	>4	R	>4	R
Erythromycin	>8	R	>8	R	>8	R	>8	R	>8	R	>8	R	>8	R
Tetracycline	>4	R	>4	R	>4	R	>4	R	>4	R	>4	R	>4	R
Chloramphenicol	>4	R	>4	R	>4	R	>4	R	>4	R	>4	R	>4	R
Fusidic acid	≥8	R	≥8	R	≥8	R	≥8	R	≥8	R	≥ 8	R	≥8	R
Imipenem	≤4	S	≤4	S	≤4	S	≤4	S	≤4	S	≤ 4	S	≤4	S
Ceftaroline	>4	R	>4	R	>4	R	>4	R	>4	R	>4	R	>4	R
Vancomycin	≤0.5	S	≤0.5	S	≤0.5	S	≤0.5	S	≤0.5	S	≤0.5	S	≤0.5	S
Amoxicillin/clavulanic	≥8/4	R	≥8/4	R	≥8/4	R	≥8/4	R	≥8/4	R	≥8/4	R	≥8/4	R

**Table 5 biomolecules-15-00266-t005:** Antibacterial activity of biosynthesized Ag-NPs against MRSA strains.

	Antimicrobial Agent	Mean of Inhibition Zone Diameter mm (Mean ± SD)	
Bacterial Strains		Ag-NPs	Oxacillin
SA-1		22 ± 1.21	0
SA-2		12 ± 2.10	0
SA-3		20 ± 1.13	0
SA-4		22 ± 2.15	0
SA-5		22 ± 2.18	0
SA-6		22 ± 1.10	0
SA-7		23 ± 1.45	0
*S. aureus* ATCC 29523		24 ± 2.10	21 ± 2.13

**Table 6 biomolecules-15-00266-t006:** MICs of biosynthesized Ag-NPs and oxacillin.

	Antimicrobial Agent	MICs µg/mL	
Bacterial Strains		Ag-NPs	Oxacillin
SA-1		12	16
SA-2		15	16
SA-3		15	32
SA-4		15	32
SA-5		15	32
SA-6		12	16
SA-7		12	16
*S. aureus* ATCC 29523		12	0.5

**Table 7 biomolecules-15-00266-t007:** Biofilm formation results in isolated MRSA.

Isolated Bacteria	Biofilm Formation									
	Before Treatment with Ag-NPs							After Treatment with Ag-NPs		
	CRA Test	MPA TEST		Biofilm Class	Biofilm Genes		MDR	CRA Test	MPA OD	MDR
		OD_C_	OD		*icaA*	*icaD*				
SA-1	+	0.007	0.234	Strong	+	+	+	-	0.010	-
SA-2	+	0.008	0.046	moderate	+	+	+	−	0.000	−
SA-3	+	0.007	0.039	moderate	+	+	+	−	0.000	−
SA-4	+	0.008	0.207	Strong	+	+	+	−	0.010	−
SA-5	+	0.008	0.011	Poor	+	−	+	−	0.000	−
SA-6	+	0.009	0.385	Strong	+	+	+	−	0.011	−
SA-7	+	0.010	0.236	Strong	+	+	+	−	0.012	−

**Table 8 biomolecules-15-00266-t008:** Checkerboard assay of oxacillin and Ag-NPs combinations against *S. aureus* SA5.

No.	MIC _Oxacillin_ + MIC _Ag-NPs_	Oxacillin + Ag-NPs (µg/mL)	FIC _Oxacillin_ + FIC _Ag-NPs_	FICi	Interpretation
1	MIC + MIC	15 + 32	1 + 1	2	indifferent
2	MIC + 1/2MIC	15 + 16	1 + 0.5	1.5	indifferent
3	MIC + 1/4MIC	15 + 8	1 + 0.25	1.25	indifferent
4	MIC + 1/8MIC	15 + 4	1 + 0.125	1.125	indifferent
5	MIC + 1/16MIC	15 + 2	1 + 0.062	1.062	indifferent
6	MIC + 1/32MIC	15 + 1	1 + 0.031	1.031	indifferent
7	MIC + 1/64MIC	15 + 0. 5	1 + 0.015	1.015	indifferent
8	MIC + 1/128MIC	15 + 0.25	1 + 0.007	1.007	indifferent
9	1/2MIC + MIC	7.5 + 32	0.5 + 1	1.5	indifferent
10	1/2MIC + 1/2MIC	7.5 + 16	0.5 + 0.5	1	additive
11	1/2MIC + 1/4MIC	7.5 + 8	0.5 + 0.25	0.75	additive
12	1/2MIC + 1/8MIC	7.5 + 4	0.5 + 0.125	0.625	additive
13	1/2MIC + 1/16MIC	7.5 + 2	0.5 + 0.062	0.562	additive
14	1/2MIC + 1/32MIC	7.5 + 1	0.5 + 0.031	0.531	additive
15	1/2MIC + 1/64MIC	7.5 + 0. 5	0.5 + 0.015	0.515	additive
16	1/2MIC + 1/128MIC	7.5 + 0.25	0.5 + 0.007	0.507	additive
17	1/4MIC + MIC	3.75 + 32	0.25 + 1	1.25	indifferent
18	1/4MIC + 1/2MIC	3.75 + 16	0.25 + 0.5	0.75	additive
19	1/4MIC + 1/4MIC	3.75 + 8	0.25 + 0.25	0.5	synergy
20	1/4MIC + 1/8MIC	3.75 + 4	0.25 + 0.125	0.375	synergy
21	1/8MIC + MIC	1.87 +32	0.125 + 1	1.125	indifferent
22	1/8MIC + 1/2MIC	1.87 + 16	0.125 + 0.5	0.625	additive
23	1/8MIC + 1/4MIC	1.87 + 4	0.125 + 0.25	0.375	synergy
24	1/16MIC + MIC	0.93 + 32	0.062 + 1	1.062	indifferent
25	1/16MIC + 1/2MIC	0.93 + 16	0.062 + 0.5	0.562	additive
26	1/32MIC + MIC	0.46 + 32	0.031 + 1	1.031	indifferent
27	1/64MIC + MIC	0.23 + 32	0.015 + 1	1.015	indifferent
28	1/128MIC + MIC	0.11 + 32	0.007 + 1	1.007	indifferent

## Data Availability

The original contributions presented in this study are included in the article/Appendix A. Further inquiries can be directed to the corresponding authors.

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
