# Peer review of "Impeding Biofilm-Forming Mediated Methicillin-Resistant Staphylococcus aureus and Virulence Genes Using a Biosynthesized Silver Nanoparticles–Antibiotic Combination"

_biomolecules, 2025, doi:10.3390/biom15020266_

Round 1

Reviewer 1 Report

Comments and Suggestions for Authors

In this Biomolecules draft, the authors described a combination method of silver nanoparticle and antibiotics to combat biofilm formation and suppressing virulence genes. The authors demonstrated MIC of 1.87 µg/mL of oxacillin and 4.0 µg/ml for Ag-NPs against MRSA when used in combination, with significant synergistic effects. The overall recommendation for this paper is that it will need major revision before acceptance. The detailed review comments are as follows:

1.      Claim around stability of Ag-NPs in suspension. The author mentioned that the zeta potential distribution is “narrow and centered” and thus the particles are stable in various biological applications. This claim is not supported by any data in the draft and the authors should conduct DLS/zeta potential study in medium that are relevant to applications, such as PBS, cell culture media, etc with different time points to demonstrate the colloidal stability.

2.      FTIR peak assignment: In FTIR assessment, the peak at 2134 cm-1 was clearly not the C-H stretch mode in methylene groups (usually around 2900). It looks more like a background signal from CO2.

3.      TEM image quality: The TEM image in figure 1 should not be distorted. It invalidates the scale bar and makes the figure very confusing.

4.      Antibacterial mechanism of Ag-NPs. The mechanism discussion in section 3.3 involved no data and all analysis was based on conclusion from citations. The author should provide direct experimental evidence and not just assume the mechanism of this work is the same as all the cited work.

5.      Claim around “Ag-NPs lead to the downregulation of biofilm-associated genes”. There is no direct gene expression data in this paper that would support the “downregulation” claim here. PCR data only showed the presence of the icaA and icaB genes, not the expression level of these genes. Citing prior work is not enough to show that the work in this draft has the same mechanism.

6.      The “Ag+” condition in time-kill assay. In section 2.9, the 5 conditions described were oxacillin, Ag-NP, 0.0MIC, 0.5MIC and 1MIC of oxacillin-Ag-NPs. However, in Figure 4 and section 3.6, the conditions were untreated, oxacillin, Ag+, 0.5MIC, 1.0MIC of oxacillin-Ag-NPs. Are “Ag+” and “Ag-NPs” the same or different?

7. There are typos in the manuscript. For example "sliver".

8. Some descriptions in results sections should be included in introduction or a separate discussion session. The author should focus more on their data and emphasize the novelty.

Author Response

We thank the reviewer for her/his constructive criticism and also for noting the importance of our study. We have revised the manuscript accordingly and all of their concerns were addressed by appropriate revision of the text. We believe the manuscript greatly improved in its rigor and presentation thanks to the reviewer’s help.

No

Comment

Response

1

Claim around stability of Ag-NPs in suspension. The author mentioned that the zeta potential distribution is “narrow and centered” and thus the particles are stable in various biological applications. This claim is not supported by any data in the draft and the authors should conduct DLS/zeta potential study in medium that are relevant to applications, such as PBS, cell culture media, etc with different time points to demonstrate the colloidal stability.

Thank you very much. We performed a DLS/zeta potential analysis in a PBS solution and confirmed the stability of Ag-NPs in suspension.

2

FTIR peak assignment: In FTIR assessment, the peak at 2134 cm1 was clearly not the C-H stretch mode in methylene groups (usually around 2900). It looks more like a background signal from CO2.

Thank you for this observation. We corrected it in the revised manuscript.

3

TEM image quality: The TEM image in figure 1 should not be distorted. It invalidates the scale bar and makes the figure very confusing

We appreciate your helpful suggestions to enhance our manuscript. We corrected it in the revised manuscript.

4

Antibacterial mechanism of Ag-NPs. The mechanism discussion in section 3.3 involved no data and all analysis was based on conclusion from citations. The author should provide direct experimental evidence and not just assume the mechanism of this work is the same as all the cited work.

Thank you very much. We propose that the antibacterial mechanisms of silver nanoparticles involve their penetration into the cell wall, disruption of the plasma membrane, and direct interaction with DNA. This interaction leads to modifications that hinder DNA replication, as indicated by the inhibition of biofilm formation and the downregulation of genes related to biofilm production

5

Claim around “Ag-NPs lead to the downregulation of biofilm-associated genes”. There is no direct gene expression data in this paper that would support the “downregulation” claim here. PCR data only showed the presence of the icaA and icaB genes, not the expression level of these genes. Citing prior work is not enough to show that the work in this draft has the same mechanism.

Thank you very much. Our findings indicate that Ag-NPs lead to the downregulation of biofilm-associated genes, particularly when these genes are not expressed during biofilm formation after treatment with Ag-NPs, as demonstrated in Table 7, not cited from previous studies

6

The “Ag+” condition in time-kill assay. In section 2.9, the 5 conditions described were oxacillin, Ag-NP, 0.0MIC, 0.5MIC and 1MIC of oxacillin-Ag-NPs. However, in Figure 4 and section 3.6, the conditions were untreated, oxacillin, Ag+, 0.5MIC, 1.0MIC of oxacillin-Ag-NPs. Are “Ag+” and “Ag-NPs” the same or different?

We appreciate your helpful suggestions to enhance our manuscript. We corrected it in the revised manuscript.

7

There are typos in the manuscript. For example "sliver".

We appreciate your helpful suggestions to enhance our manuscript. We corrected in the revised manuscript.

8

Some descriptions in results sections should be included in introduction or a separate discussion session. The author should focus more on their data and emphasize the novelty.

We appreciate your helpful suggestions to enhance our manuscript. We corrected in the revised manuscript.

Reviewer 2 Report

Comments and Suggestions for Authors

The manuscript describes antibiotic properties of the biosynthesized Silver-nanoparticles against MRSA with a completely hindered biofilm formation and without cytotoxic effects. Biogenic synthesis of the Ag-NPs with average 50 nm could uniformly and showed a good activity against the biofilm forming methicillin-resistance Staphylococcus aureus (MRSA), in a way authors have demonstrated that the Ag-NPs are formed uniformly, provided experimental supports and literature precedent with a synergistic effect with oxacillin, those are not cytotoxic. I recommend this manuscript for publication after following points are addressed by the authors.

1.        Introduction: Could author please cite ‘Methicillin-resistant Staphylococcus aureus: an overview of basic and clinical research Nature Reviews Microbiology volume 17, pages 203–218 (2019)’. This might be missed but would improve the introduction about MRSA and highlights current developments in the field.

2.        Introduction: ‘Based on information from the National Institutes of Health, 80% of recurrent microbial infections are the result of bacteria associated with biofilm formation. Treatment options that rely on nano-therapy are desperately needed, as the development of antibiotics has slowed down, and bacterial resistance has become more prevalent.’ Please cite the appropriate reference or webpage for this statement. Other than this introduction is well sum-up with literature support.

Authors have validated synthesis and demonstrated their utility in the as antibacterials, provided all the necessary experiments along with appropriate literature. I think the manuscript is ready to be published in the journal. Once, the authors add those references to the introduction.

Author Response

We thank the reviewer for her/his constructive criticism and also for noting the importance of our study. We have revised the manuscript accordingly and all of their concerns were addressed by appropriate revision of the text. We believe the manuscript greatly improved in its rigor and presentation thanks to the reviewer’s help.

No

Comment

Response

1

Introduction: Could author please cite ‘Methicillin-resistant Staphylococcus aureus: an overview of basic and clinical research Nature Reviews Microbiology volume 17, pages 203–218 (2019)’. This might be missed but would improve the introduction about MRSA and highlights current developments in the field.

We appreciate your helpful suggestions to enhance our manuscript. We revised our manuscript and rewritten some parts

2

Introduction: ‘Based on information from the National Institutes of Health, 80% of recurrent microbial infections are the result of bacteria associated with biofilm formation. Treatment options that rely on nano-therapy are desperately needed, as the development of antibiotics has slowed down, and bacterial resistance has become more prevalent.’ Please cite the appropriate reference or webpage for this statement. Other than this introduction is well sum-up with literature support.

Thank you very much. We update the references.

3

Authors have validated synthesis and demonstrated their utility in the as antibacterials, provided all the necessary experiments along with appropriate literature. I think the manuscript is ready to be published in the journal. Once, the authors add those references to the introduction.

Thank you for your valuable feedback on our manuscript

Reviewer 3 Report

Comments and Suggestions for Authors

Thanks for providing a manuscript for the study of using silver nanoparticles combined with oxacillin to against the MRSA. In this manuscript, you presented a comprehensive background introduction. The experiments were  well designed and performed. Most of your data interpretation are clear and supported. However, there are some parts of the manuscript can be improved. 

1. typo mistakes, such as:

line 67 and line 217:  add comma

line 116: "hese"

line 135: negatively charged silver ions? should be silver nanoparticles.

line 329: HFB-4 not HFP-4

2. For fig 1, the absorbance spectrum, please add the spectrum of the filtrate and the AgNO3 solution as the control samples.

3. You mentioned a few papers used the filtrate of cultivated bacterial to do the biological synthesis of nanoparticles. Could you briefly describe which components or chemicals in these culture medium play the most important role for the nanoparticle synthesis?

4. Line 35, you mentioned the concentration of Ag-NPs, could you provide the quantification method?

5. Line 42, you mentioned the average size of the NPs is 50nm, however, from Fig1C and 1E, the size of your NPs is about 20nm. 

6. In Figure4, please double check the Y-axis value, because if it is Log10 scale, there are 10^40 CFU/mL cells after 8hours incubation.  That is a huge number. Also, please correct the unit, it should be CFU, not FCU.

In addition, you cited many literatures that have demonstrated that using the combination of nanoparticles with antibiotics can improve the performance. What is the innovation of this study? Because you mentioned your results are consistent with some other works. 

Author Response

We thank the reviewer for her/his constructive criticism and also for noting the importance of our study. We have revised the manuscript accordingly and all of their concerns were addressed by appropriate revision of the text. We believe the manuscript greatly improved in its rigor and presentation thanks to the reviewer’s help.

No

Comment

Response

1

typo mistakes, such as: line 67 and line 217:  add comma line 116: "hese"

line 135: negatively charged silver ions? should be silver nanoparticles.

line 329: HFB-4 not HFP-4

We appreciate your helpful suggestions to enhance our manuscript. We corrected it in the revised manuscript.

2

For fig 1, the absorbance spectrum, please add the spectrum of the filtrate and the AgNO3 solution as the control samples

We appreciate your helpful suggestions to enhance our manuscript.  We corrected it in the revised manuscript.

3

You mentioned a few papers used the filtrate of cultivated bacterial to do the biological synthesis of nanoparticles. Could you briefly describe which components or chemicals in these culture medium play the most important role for the nanoparticle synthesis?

We appreciate your helpful suggestions to enhance our manuscript. We corrected it in the revised manuscript.

4

. Line 35, you mentioned the concentration of Ag-NPs, could you provide the quantification method?

We appreciate your helpful suggestions to enhance our manuscript. We corrected it in the revised manuscript.

The quantification of nanoparticles can be achieved through UV-Vis spectroscopy. By preparing five or more solutions of silver nanoparticles at varying concentrations, one can measure the absorbance of each sample. Subsequently, a graph can be constructed with concentration plotted on the x-axis and absorbance on the y-axis, leading to the derivation of a linear equation. This equation can then be utilized to ascertain the concentration of unknown samples.

5

. Line 42, you mentioned the average size of the NPs is 50nm, however, from Fig1C and 1E, the size of your NPs is about 20nm. 

Thank you for this observation.  We corrected it in the revised manuscript.

6

. In Figure4, please double check the Y-axis value, because if it is Log10 scale, there are 1040 CFU/mL cells after 8hours incubation.  That is a huge number. Also, please correct the unit, it should be CFU, not FCU.

Thank you for this observation.  We corrected it in the revised manuscript.

7

In addition, you cited many literatures that have demonstrated that using the combination of nanoparticles with antibiotics can improve the performance. What is the innovation of this study? Because you mentioned your results are consistent with some other works. 

Thank you for this comment. This investigation presents an innovative perspective by assessing the interaction between oxacillin and silver nanoparticles (Ag-NPs) in the fight against MRSA isolated from clinical samples and their virulence factors. .

Round 2

Reviewer 1 Report

Comments and Suggestions for Authors

Thank you for addressing most of the questions.

However, the overall mechanism is still not very convincing with current data. Gene downregulation should be tested by western blot or RT-PCR if that is the proposed mechanism. 

Author Response

We thank the reviewer for her/his constructive criticism and also for noting the importance of our study. We have revised the manuscript accordingly and all of their concerns were addressed by appropriate revision of the text. We believe the manuscript greatly improved in its rigor and presentation thanks to the reviewer’s help.

No

Comment

Response

1

However, the overall mechanism is still not very convincing with current data. Gene downregulation should be tested by western blot or RT-PCR if that is the proposed mechanism. 

We appreciate your helpful suggestions to enhance our manuscript. We tested Gene downregulation with RT-PCR in the revised manuscript as shown in figure 3.